# Glycocalyx–Sodium Interaction in Vascular Endothelium

**DOI:** 10.3390/nu15132873

**Published:** 2023-06-25

**Authors:** Lawrence Fred Sembajwe, Abdul M. Ssekandi, Agnes Namaganda, Haruna Muwonge, Josephine N. Kasolo, Robert Kalyesubula, Annettee Nakimuli, Mwesigwa Naome, Kaushik P. Patel, Sepiso K. Masenga, Annet Kirabo

**Affiliations:** 1Department of Medical Physiology, Makerere University College of Health Sciences, Kampala P.O. Box 7072, Uganda; ssekandi91@gmail.com (A.M.S.); namutoagnes@gmail.com (A.N.); harunamuwonge@gmail.com (H.M.); josephinekasolo57@gmail.com (J.N.K.); rkalyesubula@gmail.com (R.K.); 2Department of Obstetrics and Gynecology, School of Medicine, Makerere University College of Health Sciences, Kampala P.O. Box 7072, Uganda; annettee.nakimuli@gmail.com; 3Division of Clinical Pharmacology, Vanderbilt University Medical Center, Nashville, TN 37232, USA; mwesigwanaome80@gmail.com; 4Department of Cellular and Integrative Physiology, University of Nebraska Medical Center, Omaha, NE 68198, USA; kpatel@unmc.edu; 5Department of Physiological Sciences, School of Medicine and Health Sciences, Mulungushi University, Kabwe P.O. Box 80415, Zambia; sepisomasenga@gmail.com

**Keywords:** endothelial-glycocalyx, proteoglycans, syndecans, hyaluronan, glypicans, sodium chloride-salt, CVD, hypertension

## Abstract

The glycocalyx generally covers almost all cellular surfaces, where it participates in mediating cell-surface interactions with the extracellular matrix as well as with intracellular signaling molecules. The endothelial glycocalyx that covers the luminal surface mediates the interactions of endothelial cells with materials flowing in the circulating blood, including blood cells. Cardiovascular diseases (CVD) remain a major cause of morbidity and mortality around the world. The cardiovascular risk factors start by causing endothelial cell dysfunction associated with destruction or irregular maintenance of the glycocalyx, which may culminate into a full-blown cardiovascular disease. The endothelial glycocalyx plays a crucial role in shielding the cell from excessive exposure and absorption of excessive salt, which can potentially cause damage to the endothelial cells and underlying tissues of the blood vessels. So, in this mini review/commentary, we delineate and provide a concise summary of the various components of the glycocalyx, their interaction with salt, and subsequent involvement in the cardiovascular disease process. We also highlight the major components of the glycocalyx that could be used as disease biomarkers or as drug targets in the management of cardiovascular diseases.

## 1. Background

The blood vascular system consists of a network of vessels ranging from as large as the aorta, arteries, and veins to the smaller units including arterioles, venules, and capillaries [1]. With the exception of the capillaries whose wall is made up of a single layer of endothelial cells, the other vessels have a wall with several tissue layers called tunics: tunica interna, tunica media, and tunica adventitia/externa [2]. The innermost tissue layer in the wall of the blood vessels is the luminal surface layer, consisting of continuously arranged endothelial cells [3]. The single layer of endothelial cells making up the capillaries is interrupted with some fenestrations or pores in some tissues such as in the gastrointestinal tract to enable the transportation of molecules across the capillary wall [3,4,5]. The luminal surface of endothelial cells is lined with a meshwork consisting of glycoproteins, proteoglycans, and hyaluronic acid/hyaluronan (HA), forming a gel-like material known as the endothelial glycocalyx [6,7]. The main components of the endothelial glycocalyx therefore include cell surface heparan sulfate proteoglycans (HSPGs) with associated chondroitin sulfate (CS) polysaccharide chains in certain syndecans (SDCs) and hyaluronic acid/hyaluronan (HA) [8]. The HA in the glycocalyx helps to reinforce its thickness and provides it with a gel-like appearance on the surface of the endothelial cells [9]. The endothelial-associated HSPGs consist of syndecans (SDCs) and glypicans (GPCs) that are located on the cell membranes [10] and perlecan secreted into the extracellular matrix with a pericellular localization near the basement membrane [10,11].

The glycocalyx plays an important protective role on the tissues that it covers, especially the epithelial and endothelial surfaces [12]. For instance, the endothelial glycocalyx has been reported to act as a salt-buffer on the endothelial surfaces, preventing the rapid absorption of sodium chloride (NaCl) from plasma and, thus, protecting the endothelium and the underlying tissues from the damaging effects of direct exposure to excessive amounts of salt [13]. An intact glycocalyx is also essential for the normal structure or function of endothelial cells, and endothelial dysfunction is mainly associated with old age, as well as pathological conditions including inflammatory conditions, atherosclerosis, hypertension, and heart failure [14]. The extensive sulfation of the glycosaminoglycan (GAG) chains among the proteoglycans within the endothelial glycocalyx increases its negativity that allows it to bind plasma proteins such as albumin [15]. The major components of the endothelial glycocalyx that project into the vascular lumen are also involved in flow-mediated “mechanosensation-mechanotranduction” that affects nitric oxide synthesis and vasodilation [16]. In addition, the glycocalyx is involved in endothelial cell calcium dynamics as the mechanotransduction processes are mediated by the glycocalyx components are believed to either increase the calcium permeability into the cells or its release from the endoplasmic reticulum reservoirs [17]. The mechanotransduction of the luminal shear stress due to the increased blood flow into various chemical processes in the endothelial cells, involves the interaction of the glycocalyx with both membrane and cytoskeletal proteins [18].

Glycocalyx homeostasis is very important in a sense that there is constant biosynthesis and degradation of its components to maintain a balanced cell surface layer [9]. Deviations from this balance either through excessive degradation or biosynthesis of the glycocalyx has been associated with various disease states [19,20]. The excessive degradation of the glycocalyx has been linked to disease conditions including cardiovascular diseases, hypertension, diabetes, and some cancers [21]. On the other hand, increased biosynthesis of the glycocalyx components has been observed and associated with various types of cancers [21,22,23,24]. There are certain enzymes that are generally implicated in the degradation of components of the glycocalyx including: matrix metalloproteinases (MMPs), ADAM-17 (a disintegrin and metalloproteinase), and ‘sheddases’ such as secretases, heparanases, and hyaluronidases (HYAL 1to 3) [25,26,27]. Circulating levels of the glycocalyx degradation enzymes and/or levels of these components of the glycocalyx such as syndecans or HA in plasma may be used to gauge the status of the endothelial glycocalyx in various disease states [28]. Thus, prolonged cardio-pulmonary bipass in individuals undergoing cardiac surgery has been associated with increased glycocalyx degradation based on the increased levels of syndecan-1 in circulation [29].

This mini-review is intended to provide a concise summary of the literature showing the interactions between the endothelial and glycocalyx or its components with salt, strategies to reduce the damaging effects of excessive sodium chloride (NaCl) on the endothelium, and the potential use of the glycocalyx as a biomarker of the endothelial status in cardiovascular-related conditions.

## 2. Major Components of the Glycocalyx in CVD Pathophysiology

The components of the glycocalyx involved in CVD pathophysiology include membrane-bound heparan sulfate proteoglycans (such as syndecans and glypicans), CD44, and a membrane-associated polysaccharide called hyaluronic acid (or hyaluronan) (Figure 1).

### 2.1. Heparan Sulfate Proteoglycans (Syndecans and Glypicans)

The cellular biosynthesis of the HSPGs was extensively well reviewed previously [30], so we will not provide any additional details here.

#### 2.1.1. Syndecans

Syndecans belong to a family of single-pass transmembrane proteoglycans with three major domains: an ectodomain on the outside, a transmembrane domain, and a cytoplasmic domain on the inner side of the plasma membrane [31]. The ectodomain is modified with conjugated glycosaminoglycan (GAG) chains in form of either heparan sulfate (HS) or chondroitin sulfate (CS) chains or both, which enable syndecans to act as cell surface co-receptors [32]. Heparan sulfate chains are the major GAG in syndencans, but CS chains are also found on syndecans 1 and 3 [33]. The GAG chains are conjugated to the ectodomain of a core protein, expressed by any of the four mammalian genes that encode for syndecans 1 to 4 (SDCs 1 to 4) [34]. The GAG-attachment site on the ectodomain is at both ends in the case of SDC-1 and SDC-3, but it is near the terminal end for SDC-2 and SDC-4 [35]. The presence of cleavage sites on the ectodomains of syndecans renders them liable to being ‘slashed’ from the plasma membrane by ‘sheddases’ (proteolytic enzymes such as matrix metalloproteinases) and released into plasma (becoming ‘solubilized’ as soluble syndecans), which may be potentially utilized as biomarkers for various diseases [35,36,37]. The HS and CS chains are modified with unique sulfation patterns along their lengths, making them highly negatively charged and enabling them to bind with a wide range of positively charged particles including positively charged peptides (e.g., growth factors) and electrolytes including sodium ions. This configuration renders the entire cell surface glycocalyx with an overall negative charge, which serves a protective function for the epithelial and endothelial cells by binding the sodium ions and preventing the cells from damage due to excessive exposure to sodium ions.

Although no major mutations in syndecans have been reported that are associated with specific diseases, single nucleotide polymorphisms in the syndecans SDC-3 and SDC-4 are linked with lipid metabolism dysregulation [38]. The single nucleotide polymorphisms (SNPs) in the SDC-3 core-protein ectodomain (i.e., rs2282440 and rs2491132) were positively associated with obesity among Koreans [39] and with metabolic syndrome among Taiwanese people [40,41], suggesting that SDC-3 could be an important genetic modifier in the cardiovascular disease risk factors; whereas, in a cardiovascular risk study conducted among Finnish adults, the SNPs in SDC-4 were associated with obesity, hypertension, and an increased prevalence of coronary artery disease [42]. Moreover, the endothelial cells that were subjected to abnormal physical forces such as enhanced shear stress due to rapid blood flow during hypertension can cause an increased expression of SDC-4 alongside other extracellular matrix molecules [43]. Syndecans are believed to be involved in cardiac fibrosis, which is a key step in heart disease [44]. In addition, it has been demonstrated in mice that the overexpression of syndecan-4 can lead to an increased activation of Calcineurin-NFAT-dependent signaling, which exacerbates hypertrophy in an overloaded heart model [45]. Increased shedding of the endothelial cell surface proteoglycans occurs during the cardiovascular disease process, and elevated levels of some of the components of the glycocalyx such as syndecan-1 have been observed [46,47]. This has led to the suggestion that these altered levels of syndecan-1 may be potential prognostic biomarkers for patients suffering from ischemic heart disease and heart failure [48,49].

#### 2.1.2. Glypicans

Glypicans are membrane-bound or cell surface heparan sulfate proteoglycans with one or two heparan sulfate chains and a glycosylphosphatidylinositol anchor that attaches the entire molecule to the plasma membrane [50]. This group of molecules consists of six members, from glypican-1 to -6, which are expressed predominantly during embryonic development [51]. They are involved in facilitating Wnt, hedgehog, fibroblast growth factor, and bone-morphogenic proteins/peptide signaling [51,52,53]. Their GPI anchor is cleaved by the expression of an enzyme known as Notum that releases not only glypicans but also other GPI-anchored proteins from the cell surface [54]. Glypicans are a major component of the glycocalyx alongside syndecans, which undergo significant remodeling by sheer stress on the endothelial surface during atherosclerosis [55]. Glypican- 1 is the main member among the glypican family that has been confirmed to be associated with the endothelial glycocalyx, lining the luminal surface of blood vessels [30]. Glypican-1 is believed to be involved in the mechanosensation leading to shear-induced endothelial nitric oxide synthesis, which is required for improving the compliance of blood vessels through vasodilation [56].

### 2.2. Hyaluronan

Hyaluronan is a long polysaccharide of alternating glucuronic acid (GlcA) and N-acetylglucosamine (GlcNAc) sugar residues, which are un-sulfated but result in a highly negatively charged molecule [57]. It is pericellularly synthesized by a family of three pericellular enzymes known as hyaluronan synthases: from HAS-1 to -3 [58,59]. It is then degraded by a family of six hyaluronidase (HYALs) members, of which HYAL-1 and HYAL-2 are the most potent enzymes [60,61]. HYAL-2 is believed to have a role in promoting pulmonary vascular remolding and pulmonary hypertension [61]. The deficiency of HYAL-3 increases collagen deposition, promoting post-myocardial infarction fibrosis [61]. Hyaluronan is most known as an extracellular matrix molecule, but, due to its close association with various cell surface structures including its cognate receptor CD44, integrins, and proteoglycans, it contributes to the surface gel-like glycocalyx of the synthesizing cell [62]. The hyaluronan receptor-CD44 can also be cleaved from the cell surface by proteases such as ADAM-10, ADAM-17, and MMP-14 [63,64]. A delicate balance is maintained normally between hyaluronan bio-synthesis and degradation depending on the activity of HASs and HYL enzymes, which varies in different physiological and pathological conditions [62]. The hyaluronan synthesis and degradation processes are indeed crucial during cardiovascular development or angiogenesis [65,66]. Native hyaluronan is believed to inhibit angiogenesis, whereas oligosaccharides formed following hyaluronan degradation seem to promote angiogenesis by increasing endothelial cell proliferation and migration [67,68]. The interactions of hyaluronan with reactive oxygen or nitrogen species (ROS/RNS) and their consequences on tissue homeostasis are well summarized by Berdiaki et al., whereby the innate high molecular weight hyaluronan is anti-angiogenic, anti-inflammatory, and anti-oncogenic [69]. However, following degradation by the ROS/RNS, the low molecular weight oligosaccharides produced become pro-angiogenic, pro-inflammatory, and oncogenic [69]. Changes in hyaluronan synthesis and degradation have mainly been associated with cancer and inflammatory conditions [70,71,72,73]. Indeed, elevated levels of serum hyaluronan have been previously reported in patients with rheumatic mitral stenosis and pulmonary arterial thromboembolism [74]. In addition, there is also excessive abnormal angiogenesis, which is observed in the progression of cancer, diabetic retinopathy and in atherosclerosis [75,76,77,78]. Atherosclerosis is one of the major cardiovascular disease conditions involving inflammation, where hyaluronan is thought to play a role [79]. The differentially expressed hyaluronan in atheromatic aortas was found to play a role in platelet-derived growth factor (PDGF)-induced vascular smooth muscle cell proliferation and migration, which is important during atherogenesis [80]. The CD44-hyaluronan axis is believed to regulate the inflammatory process involved in atherosclerosis, making it an important potential drug target in order to bring about disease remission [81]. Indeed, overexpression of hyaluronan in the tunica media of an aorta has been demonstrated to promote the development of atherosclerosis [82]. Moreover, increased expression of some of the biosynthetic enzymes of hyaluronan, such as HAS-3, was found to increase inflammation in an atherosclerotic plaque and promote atheroprogression [83]. This suggests that hyaluronan biosynthetic enzymes such as HAS-3 could be important therapeutic targets in the fight against atherosclerosis. The inflammatory process in atherosclerosis also promotes endothelial and platelet dysfunction that exacerbate the pathogenesis of this condition [84]. More evidence showing the impairment of the endothelial glycocalyx in atherosclerosis and obesity was recently summarized by Sang Joon Ahn et al. [85].

## 3. Glycocalyx and Salt Interactions

Since the endothelial glycocalyx covers the luminal surface of the cells, where it forms a gel-like structure, it is believed to protect the endothelial cells from direct exposure to excessive NaCl salt (above 160 mEq/L) dissolved in plasma [86]. The normal sodium levels in plasma are kept within a narrow range of 135–145 mEq/L by a combination of ‘thirst’/water intake and hormonal (aldosterone-anti-diuretic/vasopressin) systems [87,88]. The GAG chains (including heparan sulfate, chondroitin sulfate, and hyaluronan) are major components of the glycocalyx and are highly negatively charged, making them attractive to the positively charged sodium ions flowing in circulation [89], Figure 2. Thus, the glycocalyx is able to play a positive role in the sodium buffering by transiently binding sodium on the luminal side of the blood vessels [90]. The glycocalyx, therefore, is a major player in buffering the intravascular sodium and stores a great amount sodium creating a hypertonic environment [91,92].

Increased sodium levels in plasma are associated with rapid degradation of the endothelial glycocalyx, which may cause endothelial dysfunction characteristic of most cardiovascular diseases [13,92,93]. Sodium overload in the blood vessels arises as a result of excessive dietary intake beyond the capacity of the kidney to excrete it, which affects the function of not only the blood vessels but also of other organs including the heart and kidneys [91,94,95].

### 3.1. Proposed Glycocalyx-Salt Interaction Mechanisms Contributing to Hypertension and Cardiovascular Disease

The mechanism underlying the rapid degradation of the endothelial glycocalyx is that excess sodium diminishes the buffering capacity of the glycocalyx leading to increased sodium reaching endothelial cells as well as reducing the repelling effect between the vascular and erythrocyte glycocalyces; this leads to corrosion of both the erythrocyte and endothelial glycocalyces, which result in endothelial activation and dysfunction [93,96,97]. Damage to the vascular glycocalyx also leads to extravasation of the excess sodium ions into the interstitial glycosaminoglycan networks where sodium disrupts the function of the glycosaminoglycans and also activates immune cells [91,98,99]. The augmented interaction between the erythrocytes and endothelial glycocalyces increases the thrombotic events [100] and the interaction between the endothelial cell and innate cells in the lumen via the adhesion molecules [101]. The entry of excess sodium through the epithelial sodium channel (ENaC) on the endothelial cells and innate immune cells (such as dendritic cells [102,103,104,105] and macrophages) activates nicotinamide adenine dinucleotide phosphate (NADPH) oxidase resulting in the generation of super oxides, peroxynitrite, and other reactive oxygen species (ROS) [106]. Normally, nitric oxide (NO) is produced to dilate blood vessels by converting L-arginine to L-citrulline and NO catalyzed by the endothelial nitric oxide synthase. The ROS react directly with the NO production pathway and inhibit the production of NO [106,107,108]. The increased oxidative stress from ROS production by the NADPH oxidase enzyme also activates the nuclear factor kappa-light-chain-enhancer of activated B (NF-κB) mediated by the NLR Family Pyrin Domain Containing 3 (NLRP3) inflammasome, leading to the production of inflammatory cytokines such as tumor necrosis factor-alpha (TNF-α), IL-1β, and IL-6 [105,109,110]. The increased production of inflammatory cytokines coupled with the reduced production of NO resulting from ROS reactions leads to the stiffening of blood vessels and endothelial activation/dysfunction that contributes to hypertension and CVDs [109]. The produced inflammatory cytokines disrupt the gap junctions between the endothelial cells and connexin hemichannels leading to increased permeability and endothelial dysfunction [111]. Further, the activated endothelial cells begin to increase their expression of adhesion molecules leading to increased rolling and adhesion of monocytes and other cells to the endothelial cells in the vasculature leading to thrombotic and increased endothelial dysfunction [112]. This is further augmented by the increased activation of NADPH oxidase that is mediated by increased salt entry via the ENaC leading to the formation of isolevuglandin (IsoLG)-protein adducts, adaptive immune activation, and secretion of inflammatory cytokines such as IL-17A, TNF-α, and interferon-gamma (IFN-γ) that contributes to the development of hypertension and exacerbates the already existing CVDs [110,113,114,115] (see Figure 3). IsoLGs are formed when ROS peroxidize arachidonic acid, a structural component of cell membranes [114]. The resulting IsoLGs bind non-covalently to lysine residues of intracellular proteins, altering their structure and function [114]. The IsoLG-protein adduct is presented to the T cells on the major histocompatibility complex (MHC) molecules of the dendritic cells as neoantigens to activate the T cells [114]. The inflammatory cytokines secreted by the T cells injure the vascular lining and cause endothelial dysfunction, and the resulting healing by fibrosis reduces the vascular lumen, stiffens the blood vessels, and accelerates atherosclerosis leading to hypertension [114]. Moreover, the delta sub-unit of the ENaC has been reported to be expressed in human arteries and its elevated levels of expression are potentially associated with hypertension [116].

The endothelial cells with excessively degraded glycocalyx from excess sodium overload are unable to produce sufficient nitric oxide due to the reduced activation of endothelial nitric oxide synthase [117,118]. Recent evidence has shown that the endothelial glycocalyx plays a role in initiating the signal transduction pathways that contribute to NO production. Specifically, Bartosch et al. demonstrated that the proteoglycan glypican-1, not syndecan-1, plays a dominant role in propagating extracellular forces into the endothelial cells to activate signal transduction for the production of NO [119]. Moreover, this plasma sodium overload contributes not only to increased loss of the endothelial glycocalyx but also to vascular inflammation, which is characteristic of various cardiovascular diseases as explained above [120]. For instance, the hypertension experienced in pre-eclampsia has been associated with increased degradation of the endothelial glycocalyx with subsequent endothelial dysfunction and vascular injury [121,122].

The endothelial glycocalyx is therefore highly sensitive to high salt intake. Not only does high salt disrupt the repulsive forces between erythrocytes and endothelial glycocalyces by saturating their buffering capacity for sodium ions, but, as mentioned earlier, high sodium also facilitates and increases the adhesion forces occurring between monocytes and endothelial surfaces leading to monocyte and endothelial cell activation that results in endothelial dysfunction and inflammation [120]. Activation of the endothelial cells increases their expression of P- and E-selectins and additional adhesion molecules that facilitate rolling and adhesion of the leukocyte to the endothelium [123,124].

It is also important to mention that the damage to the glycocalyx induced by high salt intake occurs more with chronic salt overload as opposed to acute salt loading and deterioration of the glycocalyx advances with ageing independent of the salt intake [120,125].

### 3.2. Possible Strategies for Reducing the Damaging Effects of NaCl-Salt Overload on Vascular Endothelium

Studies that explore the mechanisms associated with reducing the damaging effects of salt overload on vascular endothelium are scarce, but a few studies have reported promising results and are highlighted below.

High dietary salt intake remains a big challenge as many people in various populations around the world are still unable to stop consuming high amounts of salt due to the diverse sources of dietary salt available in common foods [126,127,128]. It is therefore understandable that efforts are being made to find alternative ways of overcoming the deleterious effects of high salt overload on human health, other than simply advising people to reduce salt intake. For example, a recent clinical trial conducted among black women aged between 20 and 60 years in the USA provided evidence showing that ‘hot yoga’ can reduce the harmful effects of salt overload on endothelial function [129]. The exact mechanism is unknown. Similarly, regular aerobic exercise has also been reported to reduce endothelin-1-mediated vasoconstriction and endothelial dysfunction in postmenopausal women, as well as in obese or overweight adults [130,131]. Replacement or substitution of NaCl salt with other forms of salt with similar ‘saltness’ taste, such as potassium chloride (KCl) and monosodium glutamate (MSG), have also been piloted by the Department of Food Science at Cornell University (New York) and been found to have relatively high acceptability among the study subjects, although with a caveat of not disclosing the specific names of the salt substitute [132]. Consumer acceptability of the salt substitute is indeed considered to be necessary in the reduction in NaCl dietary salt intake [133]. It has also been suggested that the increased intake of potassium, rather than sodium, can help to alleviate the deleterious effects of excessive sodium during the treatment of hypertension [134,135]. Other substitutes that have been used in the reduction in dietary NaCl intake include yeast extract, taste peptides, and odor compounds [136]. Since the ENaC subunits are expressed in several vascular beds including mesenteric, cerebral, and renal [137,138,139], where the increased ENaC expression is also associated with salt-sensitive hypertension by causing endothelial dysfunction or vascular smooth muscle activation [103,140], it is a therapeutic target to be blocked using drugs such as benzamil or amiloride in conjunction with other potential anti-hypertensive agents [141]. Because high extracellular salt concentration increases ENaC expression [90,142], it is logical that blocking it with amiloride helps to restore endothelial function through increased phosphorylation of endothelial nitric oxide synthase (eNOS) [138]. As the endothelial glycocalyx plays a central role in buffering against excessive salt, with its degradation preceding vascular endothelial dysfunction, preventing this damage and restoring the glycocalyx has been proposed as an important strategy in managing CVD disorders [46]. Some of the therapies that can be used to protect against degradation and help in regenerating the damaged endothelial glycocalyx include berberin, doxycycline, and sphingosine-1-phosphate (S1P) [143,144,145]. These drugs act to prevent the shedding of the glycocalyx by inhibiting the increased secretion of heparinase and MMP enzymes [145,146]. For instance, S1P binds on its receptor on the endothelial cells and increases the synthesis of syndecan-1 and heparan sulfate through phosphatidyl inositol-3 kinase (PI3K) signaling [147].

## 4. Current Diagnostic Tools to Determine Glycocalyx and Endothelial Health Status

Since the glycocalyx consists of both protein and carbohydrate components, its breakdown products include monomers or shorter/smaller molecules of each of the two large macromolecules, as well as shed syndecans and glypicans [27,148]. These include small peptides or amino acids and the breakdown products of proteins: glypicans, syndecan-1, shorter disaccharides, or individual GlcA and GlcNAc—these are sugar residues, which are the major breakdown products of the polysaccharide component of the glycocalyx [148]. The much longer HA polysaccharides are usually reduced to shorter sugar chains that are released into various body fluids including saliva, cerebrospinal fluid (CSF), urine, and blood plasma and can easily be isolated from such body fluids [149,150]. The smaller peptides or amino acids can be re-used in the synthesis of new proteins by body cells [151]. Most of the breakdown products of the polysaccharide component of the glycocalyx end up being deposited in body tissues and body fluids from which they can be measured or determined to assess the level of glycocalyx degradation as biomarkers of health or disease [148,150]. For example, elevated plasma levels of the glycocalyx components seen in sepsis-associated encephalopathy can be used as early biomarkers of cognitive impairment among sepsis patients [152]. However, Hahn et al. have summarized evidence, which seems to suggest that since glycocalyx shedding is widespread in both acute and chronic inflammatory conditions, there is a lack of adequate sensitivity or specificity for any particular disease by measuring the plasma levels of the glycocalyx components [153]. A combination of biochemical and immunological techniques such as ELISA are commonly used in the determination of the concentration of glycocalyx breakdown products [154,155]. The thickness of the glycocalyx may also be determined indirectly by using a side stream dark field imaging technique [156]. Other techniques such as atomic force microscopy and liquid chromatography with selected reaction monitoring in tandem with mass spectrometry (LC-SRM/MS) have been used to successfully determine the glycocalyx thickness disruption in both in vivo and in vitro assays [157]. Dimethylmethylene blue (GAG-DMMB) and liquid chromatography-tandem mass spectrometry (GAG-MS) assay have also been used to determine the concentration of the glycocalyx components in urine [158].

### Erythrocyte Salt Sensitivity Test

The sodium buffering capacity of both the endothelial and erythrocyte glycocalyx can be measured using a simple “salt blood test” as a way of assessing the functional capacity of the glycocalyx [159,160]. We have previously shown that a high intake of salt is associated with a damaged glycocalyx and could be potentially used to predict future CVD [161]. The salt blood test or erythrocyte salt sensitivity test has been used in several studies as a surrogate for endothelial function and should be used more often in cost limited settings [161,162]. A recent pilot study by McNally et al. demonstrated that the erythrocyte salt sensitivity test can be used as a marker for cellular salt sensitivity in hypertension [163].

## 5. Future Directions

Future research efforts should be directed toward interventions that can ameliorate the damaging effects of excessive dietary salt, since many people seem to be unable to avoid it. Additional research is needed to specifically evaluate and validate the diagnostic and prognostic potential of these glycocalyx components that are found in body fluids such as the plasma and urine of patients with cardiovascular disease.

## 6. Conclusions

Endothelial glycocalyx is important in protecting endothelial cells from direct exposure to excessive amounts of salt and helps in maintaining normal endothelial function, which is necessary in the prevention of cardiovascular diseases. The degradation of the glycocalyx is increased in certain cardiovascular diseases due to enhanced sheer force of the rapid blood flow in hypertension, inflammatory changes, and oxidative stress in atherosclerosis and, alternatively, because of the increased expression of degradative enzymes. Excessive salt due to salt overload may also lead to rapid degradation of the endothelial glycocalyx, resulting in an endothelial dysfunction and consequent cardiovascular disease. This makes dietary modification, particularly reducing sodium-salt intake or using substitutes for dietary salt, a key strategy for preserving glycocalyx integrity. The increased presence of breakdown components of the glycocalyx, such as syndecans and hyaluronan in the plasma or urine of cardiovascular disease patients can be measured and utilized as a potential prognostic biomarker for specific cardiovascular disorders.

## Figures and Tables

**Figure 1 nutrients-15-02873-f001:**
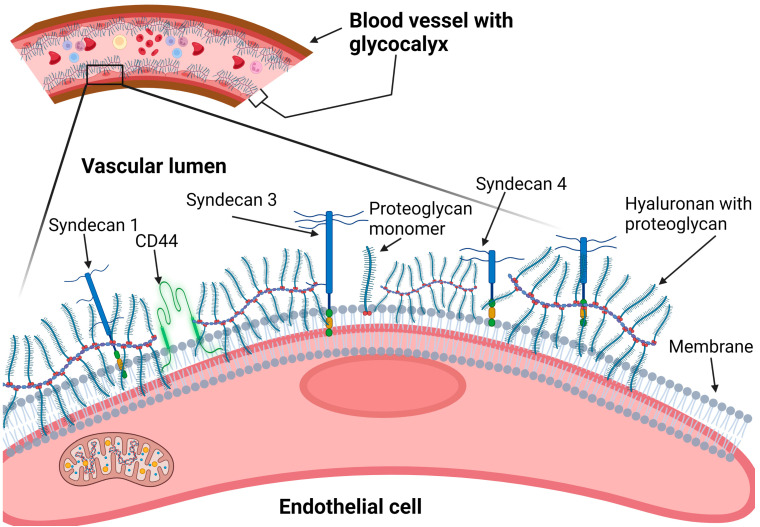
Components of the vascular endothelial cell surface glycocalyx. CD-44: cluster of differentiation-44.

**Figure 2 nutrients-15-02873-f002:**
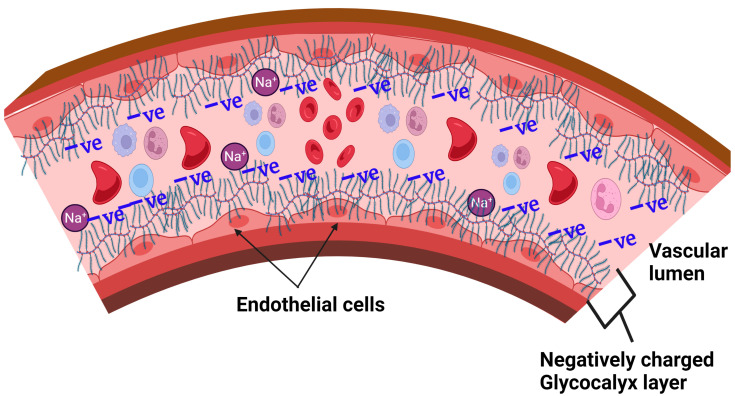
Vascular endothelial glycocalyx buffering sodium.

**Figure 3 nutrients-15-02873-f003:**
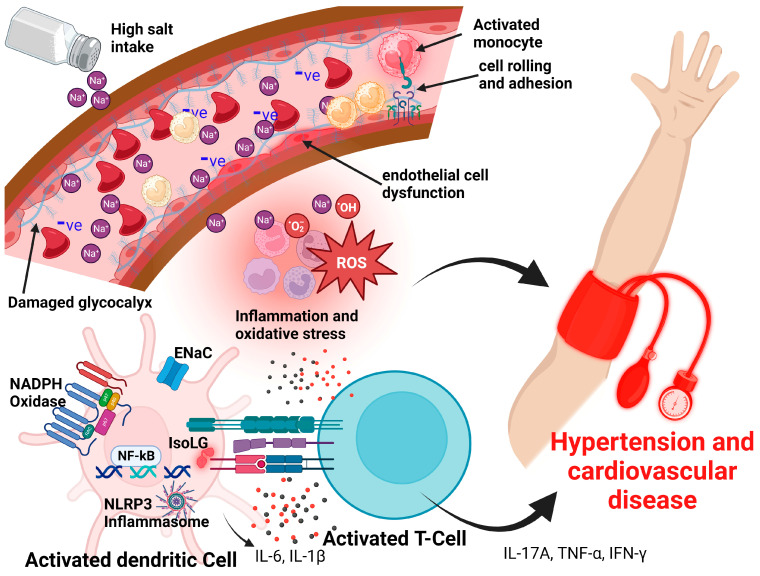
Proposed model of salt-induced damage to the glycocalyx that results in hypertension and cardiovascular disease. High intake of salt damages the glycocalyx and induces inflammation, oxidative stress, and immune activation, thus leading to the development of hypertension and cardiovascular disease. ROS, reactive oxygen species; ENaC, epithelial sodium channel; NADPH, reduced nicotinamide adenine dinucleotide phosphate; NLRP3, NLR Family Pyrin Domain Containing 3; NF-Κb, Nuclear factor kappa-light-chain-enhancer of activated B cells; IsoLGs, Islovuglandins; TNF-α, tumor necrosis factor alpha; IFN-γ, Interferon gamma.

## Data Availability

All data are contained within the manuscript.

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
