# Peer review of "Glycocalyx–Sodium Interaction in Vascular Endothelium"

_nutrients, 2023, doi:10.3390/nu15132873_

Round 1

Reviewer 1 Report

Congrats to the authors. The review deals with a meaningful, relevant and current topic. The figures are well done and didactic. I wish the authors would include one on glycocalyx degradation by proteinases such as MMPs and ADAMs. Both classes of metalloproteinases are increased in DCV. What happens to the glycocalyx? Has this been tested? In addition, doxycycline and minocycline have inhibitory activity on MMPs. Was the improvement in glycocalyx evaluated due to the inhibitory activity of MMPs? There is still a relationship between MMPs, ROS and glycocalyx degradation. Please include a discussion about this.

Author Response

Comment: Suggest to remove some content in the background …….

Response: New sentences have been used to replace the previous ones in lines: 41-49 of the first page as requested.

Comment: On the second page, a paragraph in lines 49-62, was not clear, repeated and hard to follow….

Response: This section is now removed as requested.

Comment: This statement about hyaluronan (HA) on lines 68-69 is misplaced.

Response: This has been appropriately moved to lines 52-53 in the background.

Comment: MMPs and ADAM-17 should be mentioned together not separately.

Response: MMPs and ADAM-17 has been appropriately mentioned on the second page on

lines 52-53 in the background.

Comment: What is the rationale of focusing on sodium chloride on page two?

Response: The rationale is the exposure to excessive or high sodium chloride levels and its role in CVD pathogenesis.

Comment: Correct spelling of ‘sheddases’.

Response: This has been corrected now.

Comment: On page 3, line 109 – this statement is misplaced.

Response: This statement has been appropriately relocated to lines 100-101.

Comment: On page 4, which SNPs of SDC-3 were implicated?

Response: The single nucleotide polymorphisms (SNPs) in the core protein of syndecan-3 (SDC-3) that were associated with obesity among Koreans are rs2282440 and rs2491132 and have been included on line 123, suggesting that they might be important genetic modifiers of cardiovascular disease risk factors.

Comment: Correct spelling of ‘Wnt’.

Response: This has been corrected now.

Comment: On page 4; What is the association of glypican-1 with CVD?  

Response: Glypican-1 is involved in the mechano-sensation process leading to shear-induced endothelial nitric oxide production, which is required in improving compliance of blood vessels through vasodilation. This is now included on lines 147-149.

Comment: On page 5; What is the normal range of NaCl salt in plasma?  

Response: Normal range is 135-145mEq/L; We also emphasize that we are talking about the glycocalyx protecting the endothelium from excessive NaCl salt levels above 160mEq/L on lines 194-196.

Comment: Figure 2; Whether erythrocytes play a major salt buffering role, rather than the endothelium?

Response: The endothelium is the one that plays a major role in buffering salt. However, erythrocytes are also involved in buffering sodium but to a less extent. The importance of Erythrocyte buffering capacity via its glycocalyx is for maintenance of flow by repelling endothelial glycocalyx and to also buffer additional sodium. We have now revised this section and removed part of the figure component showing erythrocyte glycocalyx to avoid the confusion and so as to emphasize that endothelial glycocalyx is the major player

Comment: What is the mechanism of rapid degradation of the glycocalyx due to exposure to excessive salt?

Response: In high salt hence high sodium, the buffering capacity of the glycocalyx for sodium is diminished. Since the glycocalyx functions to also store sodium, the excess that is not buffered extravasates into the interstitium. Moreover, the saturated buffering capacity results in disturbance of movement of cellular components that tend to move laterally. This lateral movement of cells (especially erythrocytes) close to the glycocalyx leads to erosion of the endothelial glycocalyx. This has been described in the section 3.0

Comment: Section between lines 206-226; this section needed to be the focus of the review and is sparsely supported by detail and evidence

Response: We have now included additional evidence (references) of the proposed model describing salt induced damage to the glycocalyx that results in hypertension and cardiovascular disease

Comment: On page 7, – this statement is unclear.

Response: This statement has been re-written and improved to lines 251-254.

Comment: On page 8; Table. 1; – lack of more details in the table.

Response: The table has been improved by adding a column of gene product (protein) and function

Comment: On page 10, – lack of logical flow, repeated phrases and sparce information in the manuscript.

Response: The structure of the manuscript has been improved by adding more details, deleting some sections and clarifying on others that were previously vague.

Reviewer 2 Report

This is an interesting area of research and there is a clear need to review the role of salt or circulatory Na levels in endothelial damage or in particular endothelial glycocalyx, I have a number of concerns that in my opinion prevent this review to be published in its current form. The paper touches upon the key aspects but lacks structure, logical flow, has issues with consistency and most importantly, the mechanistic detail of the interactions. At the moment, it is a weak review which does not contribute anything new to the field clearly. Detailed comments are available on the attached pdf.

Author Response

(The authors gave the same response as above.)

Reviewer 3 Report

General comments

The mini-review paper deals with an important subject: the relation between glycocalyx and the circulating sodium level. The subject is a matter of extensive study, and important review papers have appeared since 2022. Unfortunately, the present review is weak and needs an extensive comprehensive review. For example, using the glycocalyx keyword in Pubmed gives more than three hundred papers since 2022, with 139 papers since January 2023. However, the authors cited only two references in 2023. Therefore, they should update their review. In addition, if we add to glycocalyx the word endothelium or sodium salt, we get a lot of papers and reviews since 2022 that the authors did not take into consideration, such as Knezevic et al., 2023, Berdiaki et al. 2023, Almahayni and Mockl 2023, Mortazzaavi et al. 2023, Sun et al. 2023, Sculz et al. 2023, Foot et al., 2022, Patterson et al. 2022, Bkaily et al. 2022, Patik et al. 2021, ect…

Specific comments

Title: The title is misleading and should be rewritten. The authors should also specify the salt which is sodium salt. In addition, the review is focused on endothelial cells and not on the heart or vascular smooth muscle cells. Thus, the word cardiovascular should be deleted. The tile suggested is «Glycocalyx-Sodium salt interaction in vascular endothelium».

Figures:

Figs.1-3 shows blood vessels with endothelial cells covered by an external fibrous tissue. This type of blood vessel represents venules that is not implicated normally in hypertension. If the authors believe that venules are a primary target of high sodium, they should specify this and document it. If not, they should review the schematic representation of the blood vessel. In addition, it is unlikely that a cell (yellow cell) in the circulation passed through the adventitia, as indicated in Fig. 3.

The presence of ENaC in the endothelium is controversial. The authors should document the presence of ENaC in dendritic cells as showed in Fig.3. It is recommended that authors draw inspiration from the schematic representations and the role of high salt in glycocalyx from the elegant comprehensive review paper by Nijst et al., 2015, J. Am. College of Cardiol. Vol 65, and to cite this paper.

Lines 85-89; This sentence should be modified since the review focuses on endothelial cells and not on the cardiovascular system. In addition, the authors should clarify that glycocalyx destruction is due to high levels of circulating sodium and not to normal sodium as described in all papers and reviews, such as those of Nijst et al. 2015.

Lines 212- 217: As cited previously, the presence of ENaC in vascular endothelium is controversial. The authors cited the citation of a review paper by Patik et al., but not the original paper. This review argues that stiffening was prevented by amiloride, a nonspecific drug that blocks all calcium and sodium transporters. The authors should cite the original article and not only the review that cites the paper. The authors should read the original paper because, unfortunately, some authors do not cite will some articles.

The authors should have a reference for each piece of information that does not belong to them. Finally, there are 11 coauthors for a mini-review. What is the contribution of each coauthor? This should be specified.

Author Response

Comment: General comment: There is lack of more recent citations in the review

Response: This has been addressed with the addition of more recent citations from 2020 to 2023, including those suggested by the reviewer, some of the new citations included are indicated the table below:

Citation name

Citation No.

Manuscript Line No.

Mandrycky CJ.et al.,2021

1

42

Eelen G.et al.,2020

3

45

Dull RO. & Hahn RG,2022

6

50

Jin J.et al.,2021

7

50

Wang G.et al.,2020

9

53

Melrose J. 2020

11

56

Bkaily G. & Jacques D.2023

14

63

Knezevic D.et al.,2023

15

65

Foote CA.et al.,2022

16

67

Mortazavi CM.et al.,2023

17

70

Askari H.et al.,2023

18

72

Robich M.et al.,2020

29

85

Pretorius D.et al.,2022

30

102

Berdiaki A.et al.,2023

69

174 & 176

Van der Poll T.et al.,2020

84

191

Ahn SJ.et al.,2023

85

193

Paudel P.et al.,2022

114

236

Drost CC.et al.,2022

132

280

Schulz A.et al.,2023

134

284

Ermert K.et al.,2023

135

286

Baby S.et al.,2023

152

308

Hahn RG.et al.,2021

153

311

Patik JC.et al.,2021

106

334

Almahayni K.et al.,2023

164

338

Lim J.et al.,2023

171

357

Comment: The title of the manuscript was misleading and should be changed

Response: The title is changed to that suggested by the reviewer.  

Comment: Figs.1-3 shows blood vessels …. venules that are not implicated normally in hypertension. …… should specify this and document it. …. review the schematic representation of the blood vessel…… unlikely that a cell (yellow cell) in the circulation passed through the adventitia, as indicated in Fig. 3.

The presence of ENaC …. should document the presence of ENaC in dendritic …. schematic representations and the role of high salt in glycocalyx from the elegant comprehensive review paper by Nijst et al., 2015, J. Am. College of Cardiol. Vol 65, and to cite this paper.

Response: We have now modified the blood vessels in figures 1-3 to imply arteries and have removed the yellow cell. We have also added additional references of ENaC expression in dendritic cells. We have also cited the review by Nijst et al., 2015, J. Am. College of Cardiol. Vol 65

Comment: Lines 85-89; This sentence should be modified since the review focuses on endothelial cells and not on the cardiovascular system. In addition, the authors should clarify that glycocalyx destruction is due to high levels of circulating sodium and not to normal sodium as described in all papers and reviews, such as those of Nijst et al. 2015. 

Response: This statement is modified as requested on lines 87-91. Clarification has also been made in the background to emphasize excessive exposure to high sodium chloride (NaCl) salt, rather than to normal levels.

Comment: Lines 212- 217: … authors cited the citation of a review paper by Patik et al., but not the original paper. …... The authors should cite the original article and not only the review that cites the paper. …, unfortunately, some authors do not cite will some articles. 

Response: Now the original article is included and cited on line 222 by Paudel P.et al.,2022, that reported the expression of the ENaC in human beings, as well association of increased expression of ENaC with hypertension.

Comment: The authors should have a reference for each piece of information that does not belong to them.

Finally, there are 11 coauthors for a mini-review. What is the contribution of each coauthor?

Response: The whole manuscript has been revised to provide the relevant citations.

The roles of the co-authors have been stated in the section listed as “author’s contribution”, which is shown on lines: 395-400.

Round 2

Reviewer 2 Report

The authors have addressed some of the comments but while addressing comments, the structure of the review has not improved and it is very fragmented.

Some exmaples are as below:

There is still duplication of glycocalyx homeostasis in introduction and modification section later in the review. How does salt/sodium later this homestasis?

Line 228: ‘Increased sodium levels in plasma is associated with rapid degradation of the endothelial glycocalyx’ HOW? This needs to be addressed

Line 228-260: too much going on here without any clarity, needs to be rewritten with more detail

 Line 299: what does it mean by ‘modified appearance’? and in table 1, it still does not elaborate on what these modifications lead to e.g., sufatases and removal of sulfate group? Does it have any interaction with sodium? 

line 357-368 about dietary modification; this section only gives a brief information on reducing salt intake but nothing about the effect of any other dietary component e.g. polyphenols? Other interations. Reducing sal intake is a very simplistic and superficial suggestion.

Pharmacological intervention and new therapeutic targets……very confusing, lacks robust detail how is it relevant to salt/sodium?

Author Response

Point-by-point responses to the second round of reviewers’ comments regarding manuscript: Glycocalyx-Sodium interaction in vascular endothelium

We appreciate the reviewers for their thoughtful advice and guidance when reviewing our manuscript. This has improved our manuscript. We have responded to and made changes to the manuscript (track changes) accordingly. Below are the responses to the reviewer.

Thank you for your consideration

Reviewer 2 comments

Reviewer’s Comment: The authors have addressed some of the comments but while addressing comments, the structure of the review has not improved and it is very fragmented.

Response: The structure of the manuscript has been revised again and the repeated sections have been removed.

Reviewer’s Comment: Some examples are as below:

There is still duplication of glycocalyx homeostasis in introduction and modification section later in the review. How does salt/sodium later this homeostasis?

Response: Section 4.0, that was about the modification of the glycocalyx has been removed completely.

Reviewer’s Comment: Line 228: ‘Increased sodium levels in plasma is associated with rapid degradation of the endothelial glycocalyx’ HOW? This needs to be addressed

Response:  This has been addressed in lines: 218-222 as follows: “The mechanism underlying the rapid degradation of the endothelial glycocalyx is that excess sodium diminishes the buffering capacity of the glycocalyx leading to increased sodium reaching endothelial cells as well as reducing the repelling effect between the vascular and erythrocyte glycocalyces, this leads to corrosion of both the erythrocyte and endothelial glycocalyces that result in endothelial activation and dysfunction”

Reviewer’s Comment: Line 228-260: too much going on here without any clarity, needs to be rewritten with more detail

Response: This has been changed and more details have been added to the entire section as well.

Reviewer’s Comment: Line 299: what does it mean by ‘modified appearance’? and in table 1, it still does not elaborate on what these modifications lead to e.g., sufatases and removal of sulfate group? Does it have any interaction with sodium? 

Response: This section has been removed including table 1.

Reviewer’s Comment: line 357-368 about dietary modification; this section only gives a brief information on reducing salt intake but nothing about the effect of any other dietary component e.g., polyphenols? Other interactions. Reducing salt intake is a very simplistic and superficial suggestion.

Response: This section was also eliminated, instead dietary options leading to substitution of NaCl salt in diet are addressed in an added section: 3.1 Possible strategies for reducing the damaging effects of NaCl-salt overload on vascular endothelium, that is between line: 266-297

Reviewer’s Comment: Pharmacological intervention and new therapeutic targets……very confusing, lacks robust detail how is it relevant to salt/sodium?

Response: This has been removed as well.